# Quality of Life vs. Supportive Care Needs for Oral Cancer Caregivers: Are They Related?

Aira Syazleen Ahmad [1,2], Jennifer Geraldine Doss [2,3,*], Siti Mazlipah Ismail [4], Shim Chen Kiong [5], Md Arad Jelon [6], Logesvari Thangavalu [7] and Ch'ng Lay Ling [8]

1   Oral Health Program, Ministry of Health, Federal Government Administrative Centre, Putrajaya 62590, Malaysia
2   Department of Community Oral Health and Clinical Prevention, Faculty of Dentistry, University of Malaya, Kuala Lumpur 50603, Malaysia
3   Oral Cancer Research & Coordinating Centre, University of Malaya, Kuala Lumpur 50603, Malaysia
4   Department of Oral & Maxillofacial Clinical Sciences, Faculty of Dentistry, University of Malaya, Kuala Lumpur 50603, Malaysia
5   Department of Oral and Maxillofacial Surgery, Sarawak General Hospital, Kuching 93586, Malaysia
6   Department of Oral and Maxillofacial Surgery, Kuala Lumpur Hospital, Kuala Lumpur 50586, Malaysia
7   Department of Oral and Maxillofacial Surgery, National Cancer Institute, Putrajaya 62250, Malaysia
8   Department of Oral and Maxillofacial Surgery, Seberang Jaya Hospital, Perai 13700, Malaysia
*   Correspondence: jendoss@um.edu.my; Tel.: +603-7967-4805

**Abstract:** Caregivers providing care for their family members with oral cancer usually endure the caregiving burden in silence, which affects their quality of life and necessitates the need for supportive care. The aim of this study is to determine the relationship between the quality of life (QOL) of oral cancer caregivers and their supportive care needs (SCN) in Malaysia. The Malaysian versions of the Caregiver Oncology Quality of Life Questionnaire (M-CarGOQoL) and the Comprehensive Needs Assessment Tool for Cancer Caregivers (M-CNAT-C) were self-administered by 56 family caregivers of oral cancer patients from five tertiary hospitals throughout Peninsular Malaysia and Sarawak between October and December 2021. Correlation and multiple regression analyses were employed, and the significance level was set at $p < 0.05$. The mean score for the QOL of caregivers was $76.16 \pm 16.01$, with the lowest scores in the psychological well-being ($64.87 \pm 30.12$) and self-esteem ($68.64 \pm 28.29$) domains. The mean score for SCN of caregivers was $36.42 \pm 24.16$, with the highest scores in the healthcare staff ($58.44 \pm 33.80$) and information ($55.35 \pm 29.98$) domains. The correlation between QOL and SCN was moderately inversed, ($r(54) = 0.58$, $p < 0.01$). There was a significant effect of caregiving duration (<3 h/day versus >3 h/day) on the combined dependent variables (QOL and SCN), $F(2, 53) = 5.006$, $p < 0.01$, partial $\eta^2 = 0.16$. QOL and caregiving duration accounted for a significant 43% of SCN, $R^2 = 0.43$, adjusted $R^2 = 0.41$, $F(2, 53) = 20.32$, $p < 0.01$. In conclusion, oral cancer caregivers with poorer QOL have higher SCN. It is recommended that oral cancer caregivers be recognized by healthcare providers in order to deliver holistic patient care.

**Keywords:** oral cancer; caregiver; quality of life; supportive care needs





## 1. Introduction

Cancer patients and their caregivers should be treated as a unit [1–4]. Evidence reveals that as the patient's physical condition deteriorates, the caregiver's QOL deteriorates as well [3–6]. As a consequence of this, it has been suggested that the role of caregivers is significant because they contribute to the informal care of patients, as persons who help manage and support their loved ones through their journey with cancer [7].

In addition to providing clinical care to oral cancer patients, healthcare providers should be aware of the patient's surroundings, including the people who are always with them during their cancer journey. In this study, we defined caregivers as family members

living with oral cancer patients who support the patients and understand their needs in relation to daily activities during the cancer trajectory.

There is a growing body of literature that recognizes the challenges faced by caregivers through the journey of cancer caregiving. Several current studies have found that caring for cancer patients has a significant impact on caregivers' quality of life (QOL) [8–12]. Caregivers are also confronted with unmet supportive care needs (SCN) [13–17] and caregiving burden [11,12,18,19] as well. There is also mounting evidence on the caregiver's reaction towards cancer, such as grief [20], coping [21], and resilience [12]. Interestingly, researchers have also been looking into the concept of a decent death and the quality of dying as experienced by caregivers [22,23].

Recent evidence had reported that the caregiver QOL was closely related to their SCN. Caregivers who presented with poorer QOL were reported to have higher SCN, and those who were content with their lives demonstrated lower SCN [24–26]. To the best of our knowledge, the relationship between QOL and SCN in the context of oral cancer caregivers is yet to be explored. This present study intended to fill the identified gap by focusing on the QOL and SCN of oral cancer caregivers, thus, assessing the relationship between both.

## 2. Materials and Methods

Upon commencing research work, we identified the instruments to be used, which were the Comprehensive Needs Assessment Tool for Cancer Caregivers (CNAT-C) [27] to assess SCN, and Caregiver Oncology Quality of Life Questionnaire (CarGOQoL) [28] to assess QOL. To the best of our knowledge, cross-cultural adaptation of either questionnaire had yet to be done for the Malaysian population. Hence, using the forward-back method [29], the questionnaires were translated into Malay. The psychometric properties of the measures using the Rasch Measurement Model [30] concluded that the Malaysian version of both questionnaires could be extended to a larger population after a validation study on 31 oral cancer caregivers from the Oral Cancer Review Clinic at the University of Malaya was undertaken.

This cross-sectional study was conducted on caregivers of oral cancer patients in five main oral cancer referral public hospitals in Malaysia; Kuala Lumpur Hospital, National Cancer Institute, Seberang Jaya Hospital, Sarawak General Hospital, and Faculty of Dentistry, University of Malaya. We selected only one caregiver for each patient who was currently under treatment or post-treatment follow-ups through the convenience sampling technique. The inclusion criteria of the caregivers are those aged >18 years old, Malaysian, literate and able to communicate in Malay or English, and living with the patient. In our study, caregivers were defined as family members who are directly involved in helping patients with oral cancer and who are aware of their needs in regard to daily activities.

The caregivers for patients who attended follow-up appointments from January 2020 until June 2021 were identified from the oral cancer outpatient appointment list in the respective study sites. The caregivers were contacted by phone and given a brief explanation of the study's background, as well as assistance in completing the questionnaire if needed. Data were collected via postal mail from September to December 2021 (4 months), and all study participants provided informed written consent.

The questionnaire was constructed in Malay and Malaysian English, comprising three sections, Section A: Malaysian version CNAT-C (File S1: M-CNAT-C), Section B: Malaysian version CarGOQoL (File S2: M-CarGOQoL), and Section C: general information.

The M-CNAT-C consisted of 41 items within seven domains (health and psychological problems, family and social support, healthcare staff support, information, religious/spiritual support, hospital facilities and services, and practical support. With a 3-point Likert scale (1—No need help, 2—Need a little help, 3—Need a lot of help), the M-CNAT-C had a total score range of 41–123 in which higher scores denote greater SCN. The scores were linearly transformed into a range of 0–100 [27].

The M-CarGOQoL included 29 items grouped into 10 domains (psychological well-being, caregiving burden, relationship with healthcare personnel, administration and

finances, coping, physical well-being, self-esteem, leisure, social support, and private life. The M-CarGOQoL had a 3-point Likert scale (1—Never/Not at all, 2—Seldom/Sometimes, 3—Often/Always) with better QOL implied by a higher score. The scores ranged from 29–87, which include reversed items, were converted linearly into 0–100 [28]. Reverse scoring of items for M-CarGOQoL was done and reliability of internal consistency was then assessed with Cronbach alpha.

The general information section consisted of items on the caregiver's age, gender, race, financial status, employment status, location of residence, education level, number of dependents, caregiving duration, caregiving role, and their relationship with patients. Patients' details (age and intervention) were retrieved from their records in the respective study sites.

IBM SPSS Statistics 25 [31] was used for descriptive and inferential statistics. Items with more than 50% missing data were replaced using the mean series of the items. The relationship between QOL and SCN was investigated with four types of inferential analysis: (i) bivariate correlation, (ii) multivariate analysis of variance, (iii) partial correlation, and (iv) multiple regression analysis. All statistical analyses with a $p$-value $< 0.05$ were deemed statistically significant.

## 3. Results

### 3.1. Participant Characteristics (N = 56)

Excluding the oral cancer caregivers involved in the validation study, 174 caregivers were identified, of whom 90 (51.7%) agreed to participate. However, only 56 (62.2%) responded to the study. Participant characteristics are presented in Table 1. The mean age of caregivers was 43.55 years ($\pm$13.89 years) while the mean age of oral cancer patients was 59.70 years ($\pm$12.23 years). A total of 75% ($n = 42$) were female and 60.7% ($n = 34$) reported caregiving duration of more than three hours daily. Malays (46.4%), B40s (60.7%), as well as spouses and children (87.5%) were the majority groups of the caregivers.

**Table 1.** Participant demographic profiles, quality of life, and supportive care needs scores.

| Characteristics | Frequency [n (%)] | QOL Score [Mean ± SD] | Mean Difference [Test (p-Value)] | SCN Score [Mean ± SD] | Mean Difference [Test (p-Value)] |
|---|---|---|---|---|---|
| Caregivers | | | | | |
| Age | | | | | |
| 19–64 years old | 52 (92.9) | 75.08 ± 16.06 | −4.15 (0.00) * | 36.60 ± 24.95 | 0.19 (0.85) |
| ≥65 years old | 4 (7.1) | 90.21 ± 5.77 | | 34.15 ± 10.77 | |
| Gender | | | | | |
| Male | 14 (25.0) | 83.71 ± 10.97 | 2.58 (0.01) ** | 35.19 ± 22.30 | −0.22 (0.83) |
| Female | 42 (75.0) | 73.64 ± 16.73 | | 36.83 ± 25.00 | |
| Race | | | | | |
| Malay | 26 (46.4) | 78.24 ± 15.58 | 1.51 (0.68) | 33.40 ± 22.08 | 1.16 (0.34) |
| Chinese | 14 (25.0) | 71.34 ± 16.23 | | 42.68 ± 26.80 | |
| Indian | 13 (23.2) | 76.13 ± 17.51 | | 40.06 ± 25.72 | |
| Other Bumiputera | 3 (5.4) | 80.65 ± 14.68 | | 17.69 ± 17.81 | |
| Financial status | | | | | |
| Bottom 40% of the Malaysian household income (B40) | 34 (60.7) | 72.47 ± 17.60 | 2.77 (0.07) | 38.16 ± 25.24 | 0.43 (0.66) |
| Middle 40% of the Malaysian household income (M40) | 20 (35.7) | 82.71 ± 11.29 | | 34.79 ± 22.37 | |
| Top 20% of the Malaysian household income (T20) | 2 (3.6) | 73.28 ± 10.97 | | 23.17 ± 32.77 | |

**Table 1.** *Cont.*

| Characteristics | Frequency [*n* (%)] | QOL Score [Mean ± SD] | Mean Difference [Test (*p*-Value)] | SCN Score [Mean ± SD] | Mean Difference [Test (*p*-Value)] |
|---|---|---|---|---|---|
| **Employment status** | | | | | |
| Government | 5 (8.9) | 85.15 ± 5.69 | | 25.85 ± 22.34 | |
| Private | 19 (33.9) | 70.65 ± 13.92 | | 39.60 ± 23.23 | |
| Self-employed | 5 (8.9) | 79.03 ± 14.62 | 2.30 (0.07) | 43.90 ± 21.78 | 0.54 (0.71) |
| Retired | 6 (10.7) | 89.54 ± 5.17 | | 29.58 ± 8.73 | |
| Unemployed | 21 (37.5) | 74.49 ± 19.16 | | 36.24 ± 28.99 | |
| **Location** | | | | | |
| Urban | 36 (64.3) | 77.14 ± 15.09 | 0.61 (0.54) | 34.77 ± 23.73 | −0.68 (0.50) |
| Rural | 20 (35.7) | 74.39 ± 17.83 | | 39.39 ± 25.26 | |
| **Education** | | | | | |
| PhD/Master/Degree | 19 (33.9) | 77.46 ± 15.16 | | 43.26 ± 25.74 | |
| Malaysian Higher School Certificate (STPM)/Diploma or equal | 12 (21.4) | 78.36 ± 14.73 | 0.26 (0.93) | 29.22 ± 20.35 | |
| Vocational/Certificate or equal | 3 (5.4) | 69.70 ± 14.57 | | 41.87 ± 18.51 | 0.82 (0.54) |
| Malaysian Certificate of Education (SPM) or equal | 15 (26.8) | 74.84 ± 18.74 | | 33.90 ± 22.63 | |
| Lower Secondary Assessment (PMR) or equal | 4 (7.1) | 71.20 ± 21.88 | | 41.77 ± 29.44 | |
| Primary school/No formal education | 3 (5.4) | 78.74 ± 14.86 | | 21.95 ± 36.97 | |
| **Number of dependents** | | | | | |
| None | 7 (12.5) | 73.68 ± 23.54 | | 35.89 ± 22.73 | |
| One | 12 (21.4) | 76.02 ± 11.14 | 0.10 (0.91) | 35.22 ± 23.07 | 0.02 (0.98) |
| Two or more | 37 (66.1) | 76.67 ± 16.13 | | 36.92 ± 25.35 | |
| **Duration of caregiving** | | | | | |
| <3 h/day | 22 (39.3) | 77.58 ± 14.25 | 0.53 (0.60) | 25.64 ± 23.62 | −2.88 (0.00) * |
| ≥3 h/day | 34 (60.7) | 75.23 ± 17.20 | | 43.40 ± 22.15 | |
| **Role of caregiving** | | | | | |
| Shared | 36 (64.3) | 75.06 ± 14.82 | −0.68 (0.50) | 38.96 ± 21.21 | 0.97 (0.34) |
| Single | 20 (35.7) | 78.13 ± 18.21 | | 31.86 ± 28.76 | |
| **Relationship with patients** | | | | | |
| Spouse | 21 (37.5) | 79.92 ± 15.81 | | 28.95 ± 22.42 | |
| Parent | 2 (3.6) | 78.50 ± 18.21 | | 40.24 ± 1.72 | |
| Sibling | 3 (5.4) | 78.29 ± 11.21 | 2.83 (0.59) | 36.18 ± 31.62 | 0.95 (0.47) |
| Child | 28 (50.0) | 73.15 ± 17.01 | | 42.03 ± 25.76 | |
| Others | 2 (3.6) | 73.28 ± 10.97 | | 32.93 ± 5.17 | |
| **Patients** | | | | | |
| **Age** | | | | | |
| 22–64 years old | 36 (64.3) | 76.39 ± 15.38 | 0.15 (0.88) | 31.69 ± 23.88 | −2.05 (0.05) ** |
| ≥65 years old | 20 (35.7) | 75.73 ± 17.50 | | 44.94 ± 22.82 | |
| **Intervention** | | | | | |
| No active treatment | 3 (5.4) | 73.63 ± 27.02 | | 33.33 ± 27.82 | |
| Surgical | 20 (35.7) | 81.59 ± 12.51 | 1.30 (0.29) | 30.00 ± 18.82 | 1.09 (0.36) |
| Surgical + adjuvant | 29 (51.8) | 72.59 ± 16.93 | | 42.01 ± 26.89 | |
| Non-surgical | 4 (7.1) | 76.76 ± 14.89 | | 30.34 ± 23.66 | |

*N* = 56, * *p* < 0.01, ** *p* < 0.05

### 3.2. Oral Cancer Caregiver QOL

Figure 1 illustrates the total mean score for caregivers' QOL as 76.16 ± 16.01, which was at the upper score range. The total score range included a minimum score of 39.76 and maximum score of 98.28. In ascending order, the mean scores for M-CarGOoL domains were psychological well-being = 64.87 ± 30.12, self-esteem = 68.64 ± 28.29, relationship with healthcare staff = 71.16 ± 30.75, private life = 76.49 ± 15.45, physical well-being = 78.64 ± 27.25, leisure = 78.88 ± 26.78, administration and finances = 79.09 ± 22.30, coping = 80.61 ± 23.73, caregiving burden = 82.73 ± 25.73, and social support = 81.51 ± 29.49.

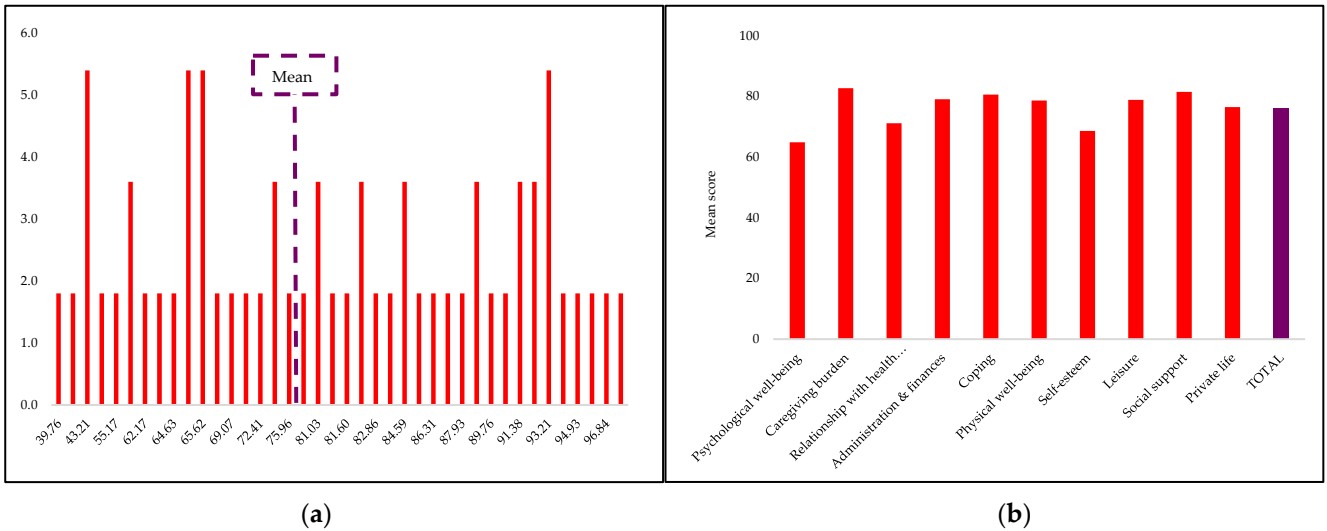

(**a**)  (**b**)

**Figure 1.** Oral cancer caregivers' QOL scores. (**a**) Frequency of QOL scores (%); (**b**) M-CarGOQoL domains mean scores.

### 3.3. Oral Cancer Caregiver SCN

As shown in Figure 2, the total mean score of caregivers' SCN was 36.42 ± 24.16, at the lower end of the score range, with a minimum score of 0.00 and maximum score of 95.12. In descending order, the mean scores for M-CNAT-C domains were healthcare staff support = 58.44 ± 33.80, information = 55.35 ± 29.98, religious/spiritual support = 36.16 ± 37.21, hospital facilities and services = 29.32 ± 29.30, practical support = 26.79 ± 29.16, health and psychological problems = 17.26 ± 24.66, and family and social support = 14.11 ± 22.06.

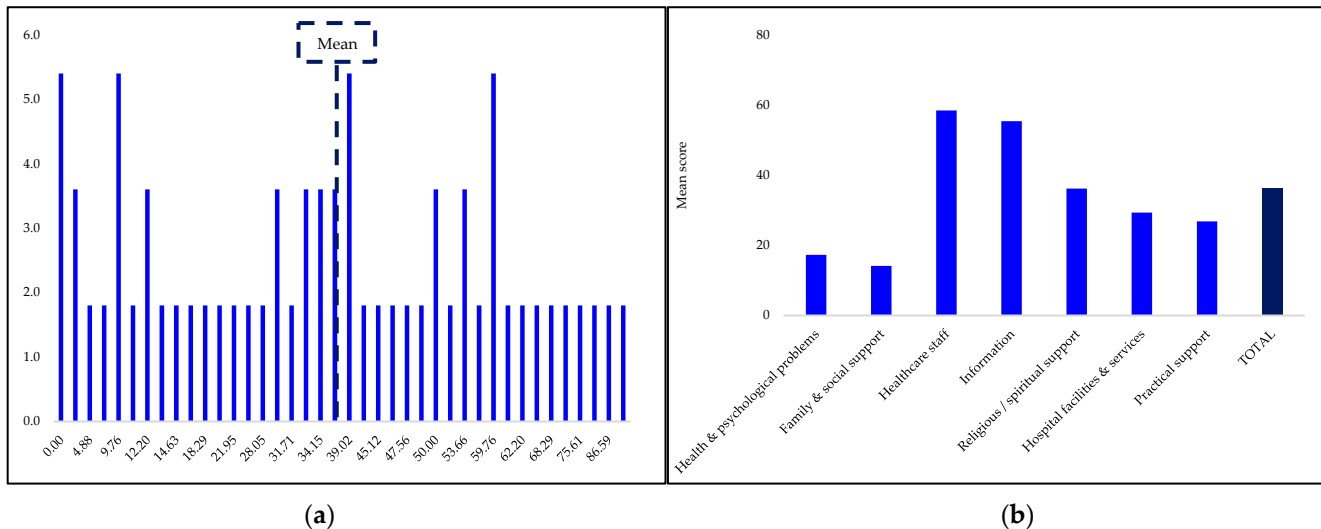

(**a**)  (**b**)

**Figure 2.** Oral cancer caregivers' SCN scores. (**a**) Frequency of SCN scores (%); (**b**) M-CNAT-C domains mean scores.

*3.4. Relationship between Oral Cancer Caregiver QOL and SCN*

3.4.1. Correlation between QOL and SCN

The bivariate correlation between QOL and SCN was negative and moderate, $r(54) = 0.58$, $p < 0.01$. Further correlation analyses were performed to assess the relationship between QOL score and M-CNAT-C domains (Table 2).

**Table 2.** Correlations between total M-CarGOQoL and M-CNAT-C domain scores.

| M-CNAT-C | Correlations | Sig. (2-Tailed) |
|:---:|:---:|:---:|
| **Overall** | −0.58 * | 0.00 |
| Health and psychological problems | −0.59 * | 0.00 |
| Family and social support | −0.49 * | 0.00 |
| Healthcare staff | −0.37 * | 0.01 |
| Information | −0.47 * | 0.00 |
| Religious/spiritual | −0.39 * | 0.00 |
| Hospital facilities and services | −0.38 * | 0.00 |
| Practical support | −0.51 * | 0.00 |

$N = 56$, * $p < 0.01$

3.4.2. Caregiving Duration as a Confounding Factor of Oral Cancer Caregiver's QOL and SCN

Multivariate analysis of variance showed that there was a significant effect of caregiving duration (<3 h/day versus >3 h/day) on the combined dependent variables (QOL and SCN), $F(2, 53) = 5.006$, $p < 0.05$, partial $\eta^2 = 0.16$. The findings were not significant for other caregiver characteristics. The oral cancer caregiver's SCN was statistically significant at a Bonferroni adjusted alpha level of 0.03, $F(1, 54) = 8.16$, $p < 0.05$, partial $\eta^2 = 0.13$. The oral cancer caregivers with caregiving duration of >3 h/day reported significantly higher SCN (mean = 43.40) than those with caregiving duration of <3 h/day (mean = 25.64).

Partial correlation was used to assess the linear relationship between QOL and SCN of oral cancer caregivers, after controlling for caregiving duration. The partial correlation was statistically significant, $r(54) = 0.590$, $p < 0.01$. An inverse moderate relationship between QOL and SCN was evident after controlling for caregiving duration.

3.4.3. Prediction of Oral Cancer Caregiver SCN

QOL and caregiving duration accounted for a significant 43% of SCN, $R^2 = 0.43$, adjusted $R^2 = 0.41$, $F(2, 53) = 20.32$, $p < 0.01$. This indicates that about 43% of the variation in the SCN score is explained by the QOL and caregiving duration. Table 3 shows the unstandardized ($B$), standardized ($\beta$) regression coefficients, and squared semi-partial correlations ($sr^2$) for each predictor in the regression model. From the $sr^2$ value, it was evident that around 35% variance in QOL could be uniquely attributed to SCN, higher than caregiving duration, which is only attributed by 15% variance of SCN. The oral cancer caregiver's QOL score ($\beta = 0.55$, $p < 0.01$) was a significantly better predictor of their SCN compared to caregiving duration ($\beta = 0.32$, $p < 0.05$) with $R^2 = 0.43$.

**Table 3.** Unstandardized (*B*), standardized (*β*) regression coefficients, and squared semi-partial correlations (*sr*$^2$) for each predictor in the regression model.

| Variable | *B* [95% *CI*] | *β* | *sr*$^2$ |
|---|---|---|---|
| (constant) | 74.40 | | |
| QOL | −0.83 [−1.15, −0.52] | −0.55 * | 0.35 |
| Caregiving duration | 15.81 [5.62, 26.00] | 0.32 ** | 0.15 |

*N* = 56, *CI* = confidence interval, * *p* < 0.01, ** *p* < 0.05

## 4. Discussion

The oral cancer caregiver QOL in the present study was in the moderate range and is in tandem with earlier studies assessing the QOL of oncology caregivers using the adapted Minaya's CarGOQoL [32–34]. Since the mean scores for each domain were lower than the total mean score, it is evident that psychological well-being, self-esteem, and relationship with healthcare personnel were the most affected QOL issues for oral cancer caregivers in Malaysia.

Previous studies had brought to light the issues of psychological implications on the cancer caregiver's QOL [18,34,35]. Goswami and Gupta (2020) revealed that possibly despite adapting and changing their daily routine, caregivers suffered from psychological impact [10]. Deeper insight into the psychological impact of caregiving among these caregivers demonstrated that they experienced distress, anxiety, fear, and uncertainty while caring for and supporting the family members who were ill and watching their loved ones in pain made them sad and depressed [36]. However, the findings of the present study contradicted the conclusions of another study using the same questionnaire (CarGOQoL) in which the most significant issue that affects caregivers was related to leisure, while self-esteem had the least impact on caregivers [34]. Respondents in the present study perceived greater satisfaction about leisure, suggesting that they were more content with the free time they had while caring for the patients.

The total mean SCN score for caregivers in the current study was higher than earlier research that used the adapted Shin's CNAT-C [13]. This may indicate that the oral cancer caregivers of the country require further assistance in providing care. It was evident that the healthcare staff assistance and need for information were the two types of support that oral cancer caregivers most critically require since the mean scores for both domains were greater than the total mean score. These findings were further reinforced by other studies that also concluded that caregivers absolutely needed support related to healthcare staff [16,25,37–39] and information [13,14,16,38,40] from healthcare providers.

Apart from family and social support, health and psychological support were the least needed forms of assistance, contrary to the findings of a recent study [14]. Contradictory to the fact that psychological well-being had the most negative effects on caregiver QOL in the present study, there was surprisingly little need for psychological support. This could be possibly be due to their effective coping behaviors or perhaps the fear of stigmatization by others in acknowledging they actually needed psychological help. In addition, the caregivers also recognized that their needs for religious/spiritual support, hospital facilities and services, and practical support had been met satisfactorily, in contrast to prior studies that revealed a strong need for practical support [14] and hospital facilities and services [16].

An association between caregiver QOL and SCN had been established in earlier research [24–26,37,39]. The bivariate correlation analysis postulated that oral cancer caregivers with poorer QOL demonstrated higher SCN. In accordance with earlier studies, further bivariate correlation analysis between the QOL and M-CNAT-C domains revealed inverse correlations for all domains [25,26,39]. To assist caregivers in achieving a satisfactory QOL, it is critical to have insight into their unmet needs and sustaining caregiver QOL is crucial in the delivery of high-quality care. Consequently, initiatives to address unmet needs and enhance the QOL of these affected oral cancer caregivers can result in better treatment outcomes for patients with oral cancer as well [41,42].

We had hypothesized that some caregiver characteristics may influence both QOL and SCN. The significant effect of caregiving duration on the combined dependent variables showed that caregiving duration had a confounding effect on both QOL and SCN. These results reinforced previous findings that the time spent by caregivers was a crucial component of cancer caregiving [4,43–46]. Further partial correlation analysis revealed higher correlation between caregiver QOL and SCN without the caregiving duration effect, which served to further support the strong relationship between QOL and SCN.

Provided the caregiver QOL level and duration of caregiving are known, healthcare personnel may anticipate the caregiver SCN with the aid of the predictors discovered in the present study. In tandem with a previous study, an increase in SCN levels among caregivers outweighed the deterioration in QOL [26]. The present study has shown that caregiver QOL appeared to be a better determinant of their SCN compared to their caregiving duration. As a response, relevant stakeholders should adopt proactive intervention strategies to safeguard caregiver QOL, so that they remain motivated and their level of SCN is satisfactory. Needless to say, integrated management of caregivers simultaneously with oral cancer patients is one such strategy to be considered as part of holistic care. One initiative would be to introduce a routine QOL assessment for all accompanying caregivers of oral cancer patients in all oral and maxillofacial specialist clinics in order to address their unmet supportive needs effectively.

The evidence obtained in the current study might assist the healthcare providers in holistic provision of care, since numerous existing studies had shown that the QOL of caregivers and patients were related [8–12,43]. Early intervention for caregivers with poor QOL can indirectly help improve the overall well-being of oral cancer patients. The healthcare providers could find ways to support a caregiver whose QOL is known to be low by addressing the significant M-CNAT-C domains. The findings of this study urge healthcare practitioners to identify cancer caregivers who are most likely to experience hardship and refer them to the right sources of care.

Although caregiving duration was found to be significantly related to SCN, QOL was revealed to be a superior predictor. This leads to the idea that in order to support caregivers, it is vital that their QOL be assessed. This is especially important given that the current study findings indicate that caregivers most frequent SCN is for healthcare staff and information support. Therefore, in order for healthcare professionals to customize an intervention program to assist caregivers for oral cancer patients, evidence in this study could be helpful.

The present study had some strengths. Firstly, it is a pioneer study in Malaysia to examine the QOL and SCN of oral cancer caregivers. The findings of this study provide baseline data which can inform future planning, decision-making, and policies for caregivers, particularly for oral cancer patients in Malaysia. Secondly, the five study sites selected for the field study were also one of its strengths as these were main tertiary oral cancer centers throughout Malaysia, thus reflecting managed oral cancer patients from across the country, including urban and rural areas. Thirdly, the response rate of this study was reasonably high at 62.2% compared to a prior study that investigated the relationship between family caregiver SCN and QOL using the same data collection approach, which rendered only 33% respondents [26].

In contrast, the relatively small sample ($N = 56$) was one of the study limitations, probably due to the changes caused by the Movement Control Order (MCO) during the COVID-19 pandemic. Since everyone's movement was restricted, many oral cancer patients cancelled or postponed their appointments due to fear and anxiety of contracting the disease in public places such as hospitals. This pandemic created mental health concerns in addition to physical ones, according to various local studies [47–49]. The data collection technique using postal mail in this study may have compromised sample size too. However, considering the older age profile of the target group, this method of data collection was deemed more appropriate than an online survey. Furthermore, the elderly (>65 years old) in Malaysia were reported to have an online usage rate of only 2.0% [50], and it was expected

that some caregivers would not respond owing to Internet challenges. Another limitation in the present study is that although the sample was considered nationally representative of caregivers for oral cancer patients, it may not have included those who sought treatment elsewhere, such as private hospitals or other health institutions.

## 5. Conclusions

The QOL and SCN of caregivers for oral cancer patients were found to be related and significantly inversed, implying that caregivers with poorer QOL had higher SCN. Caregiver QOL and SCN differed between caregiving duration of less than 3 h/day and more than 3 h/day, which implies that caregiving duration confounded the relationship between the QOL and SCN. Without the effect of caregiving duration, the inverse correlation between QOL and SCN was significantly stronger. In addition, between the QOL and caregiving duration, the caregiver QOL was a better predictor of their SCN.

In general, this study has contributed towards the advancement of knowledge in the field of caregivers for oral cancer patients, particularly in terms of their QOL and SCN. With all the evidence acquired in this study, the most essential step that needs to be undertaken is to raise awareness among all relevant stakeholders that oral cancer caregivers in Malaysia too require proper attention in terms of their QOL and SCN. A dedicated team in the hospital could be formed in order to implement support programs for oral cancer caregivers. The intervention program for the support of oral cancer caregivers could be planned through a collaborative effort of the oral surgeons, nurses, and counsellors. Soft skill training for healthcare personnel, particularly those who work directly with oral cancer caregivers in the hospitals, is much needed to increase the quality of service delivery to oral cancer caregivers in Malaysia.

**Supplementary Materials:** The following supporting information can be downloaded at: https://www.mdpi.com/article/10.3390/curroncol30020134/s1, File S1: M-CNAT-C; File S2: M-CarGOQoL.

**Author Contributions:** Conceptualization, A.S.A., J.G.D. and S.M.I.; data curation, A.S.A.; formal analysis, A.S.A.; funding acquisition, A.S.A. and J.G.D.; investigation, A.S.A., S.C.K., M.A.J., L.T. and C.L.L.; methodology, A.S.A., J.G.D. and S.M.I.; project administration, A.S.A. and J.G.D.; supervision, J.G.D. and S.M.I.; validation, J.G.D. and S.M.I.; writing—original draft, A.S.A.; writing—review and editing, A.S.A., J.G.D., S.M.I., S.C.K., M.A.J., L.T. and C.L.L. All authors have read and agreed to the published version of the manuscript.

**Funding:** This research was funded by Dental Postgraduate Research Grant Faculty of Dentistry University of Malaya, grant number DPRG/05/21.

**Institutional Review Board Statement:** The study was conducted in accordance with the Declaration of Helsinki, and approved by Medical Ethics Committee of Faculty of Dentistry University of Malaya (protocol code DF CO2013/0074 [P]; 24 June 2020–01 November 2022) and Medical Research & Ethics Committee of Ministry of Health Malaysia (protocol code NMRR-20-3197-56291; 1 June 2021–31 May 2022).

**Informed Consent Statement:** Informed consent was obtained from all subjects involved in the study.

**Data Availability Statement:** Not applicable.

**Acknowledgments:** The Oral Health Program, Ministry of Health Malaysia was made aware of this study and provided support for its conduct.

**Conflicts of Interest:** The authors declare no conflict of interest.

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
