# Peer review of "Quality of Life vs. Supportive Care Needs for Oral Cancer Caregivers: Are They Related?"

_curroncol, doi:10.3390/curroncol30020134_

Round 1

Reviewer 1 Report

I congratulate the authors for addressing an issue of great current interest and one that needs to be addressed. Following minor revisions are suggested:

1. The period about 56 responders among 174 identified caregivers in M&M (lines 88-91; "With exclusion of oral.......62% respondents to the study.") should be moved to the 'results' section. 

2. In the table 1, it is necessary to add the details of the acronyms shown (e.g. B40/M40/T40 or STPM/SPM/PMR).

Author Response

Dear reviewer,

Thank you.

Reviewer 2 Report

Manuscript Title: QUALITY OF LIFE vs SUPPORTIVE CARE NEEDS FOR ORAL CANCER CAREGIVERS: ARE THEY RELATED?

Thank you for asking me to assess the above-titled manuscript.   Comments for the Authors   GENERAL

This review attempts to determine the relationship between the quality of life (QOL) of oral cancer caregivers’ and their supportive care needs (SCN) in Malaysia.  

Here are my observations on this study/paper:

Abstract

Entirely appropriate.
Aims and objectives of the review are clear

Introduction

·       Lack essential details.

·       There are a number of up to date references which should be included and omit the entire outdated one.

·        Aims and objective of the study are clearly stated.

Material and Methods

·       Appropriate.

Results

Appropriate

Discussion

  • Adequate

Conclusion

·       The conclusions are justified and supported by the results.

References

·       Some of the references do not conform to the journal style.

Overall Conclusion:
1.     Originality of the manuscript: Data original, but the concept is not new
2.     Scientific merit: Average

3.     Organization: Excellent
4.     Tables necessity: Adequate
5.     Clarity: Excellent.
6.     Adequate support of the conclusions: Yes

Author Response

Dear reviewer,

Thank you.

Reviewer 3 Report

This is an interesting study to discuss the relationship between QOL and SCN of the caregiver. I think it is an important issue, however, several points, especially the Material and Method sections, need to be clearer.

1.      In the material and Methods section, please adjust the writing order according to the following paragraph. Because it is difficult for the reader to read even if the authors have covered most of the information

            i.                Study design and patients’ population: the inclusion and exclusion criteria

          ii.                The definition of caregiver: please mention the definition clearly, and line 88-91, “ With the exclusion of the oral cancer caregivers involved in the validation study and using convenience sampling technique….”, what is the sentence mean, is it considered to remove to the results section.

        iii.                The general clinical information

        iv.                The Questionnaire used in the study: included the quality of life and supportive care need score

          v.                Statistical analysis

2.      In table 1 and line 126, please mention the full name of the financial status, including B40, M40, and T40

3.      In Table 1, the column of frequency, could the author change the order, write the number of patients first, and write the percentage later; but not the percentage first then the patient number 

4.      In Table1, 3 patients were treated by “conservative”, what is it mean, and what is the difference from non-surgical, does it have a clear definition

Author Response

Dear reviewer,

Thank you.

Round 2

Reviewer 3 Report

I think the manuscript is ready to be accepted